# Cryptic bioactivity capacitated by synthetic hybrid plant peptides

Yuki Hirakawa[1], Hidefumi Shinohara[2], Kai Welke[1,3], Stephan Irle[1,3], Yoshikatsu Matsubayashi[2], Keiko U. Torii[1,2,4,5] & Naoyuki Uchida[1,2]

Evolution often diversifies a peptide hormone family into multiple subfamilies, which exert distinct activities by exclusive interaction with specific receptors. Here we show that systematic swapping of pre-existing variation in a subfamily of plant CLE peptide hormones leads to a synthetic bifunctional peptide that exerts activities beyond the original subfamily by interacting with multiple receptors. This approach provides new insights into the complexity and specificity of peptide signalling.

[1] Institute of Transformative Bio-Molecules (WPI-ITbM), Nagoya University, Furo-cho, Chikusa-ku, Nagoya 464-8601, Japan. [2] Division of Biological Science, Graduate School of Science, Nagoya University, Chikusa, Nagoya 464-8602, Japan. [3] Department of Chemistry, Graduate School of Science, Nagoya University, Chikusa, Nagoya 464-8602, Japan. [4] Department of Biology, University of Washington, Seattle, Washington 98195, USA. [5] Howard Hughes Medical Institute, University of Washington, Seattle, Washington 98195, USA. Correspondence and requests for materials should be addressed to K.U.T. (email: ktorii@u.washington.edu) or to N.U. (email: uchinao@itbm.nagoya-u.ac.jp).

Stem cell activities in plants are controlled by intercellular signalling through CLE (CLV3/ESR) family of peptide hormones. Precursor proteins encoded by *CLE* genes are posttranslationally processed into mature 12 or 13 amino-acid CLE peptides[1–3]. Among 32 genes in the *Arabidopsis thaliana CLE* family, different sets of members are expressed in the three types of stem cell tissues: shoot, root and vascular meristems, regulating the homeostasis of stem cell populations[4]. Notably, the *CLE* genes acting for the shoot and root meristems (*CLV3* and *CLE40*, respectively) are functionally exchangeable as revealed by a promoter-swapping analysis[5,6] and indeed chemically synthesized 12 amino-acid CLV3 acts on both tissues; its overdose diminishes the growth of shoot and root[3]. In contrast, CLE41 peptide, also called tracheary element differentiation inhibitory factor (TDIF), promotes the stem cell activity in the vascular meristem without affecting stem cells in the shoot and root[1,7,8]. CLV1 and TDR are transmembrane receptors for CLV3 and CLE41/TDIF, respectively, and the difference between their bioactivities lies in the specificity in ligand-receptor interaction[7,9,10], although the structural basis of the specificity has not been fully understood[11–13].

## Results

**Identification of a bifunctional CLE peptide**. CLV3-like activities have also been reported in other 18 CLE peptides. Some of them, including CLE25, are more effective than CLV3 in the root-shortening assay[14]. We analysed the structure–activity relationship between CLV3 (Fig. 1a, peptide 1) and CLE25 (Fig. 1a, peptide 2) by swapping the residues at 2nd, 5th, 10th and 12th positions (Fig. 1a, peptide 3–16). Both 1 μM CLV3 and CLE25, but not CLE41 (Fig. 1a, peptide 17), showed root-shortening activities with quantitatively different degrees (Fig. 1b). Although all the chimera peptides shortened the root, only the peptide 15, hereafter called KIN, showed a CLE25-like hyperactivity, indicating that the $K^{2nd}$, $I^{10th}$ and $N^{12th}$ are responsible for the strong activity of CLE25 (Fig. 1b). A dose–response assay supported the finding that KIN acts in a similar manner to CLE25 rather than CLV3; CLV3 attenuates the root growth in a lower concentration than CLE25 and KIN, whereas the effect of KIN was comparable to that of CLE25 (Supplementary Fig. 1). The size of the root apical meristem (RAM) was reduced in CLV3-, CLE25- and KIN-treated plants, which was not observed in CLE41-treated plants (Fig. 1c). We further examined the effect of KIN on the growth of the shoot apical meristem (SAM). The dome-shaped SAM disappeared after a 10-day treatment of 10 μM CLV3, which promotes differentiation of stem cells[2,15] (Fig. 1d). KIN also showed a SAM-consuming activity, whereas CLE41 did not (Fig. 1d). In the KIN-treated plants, a small residual SAM was observed, suggesting that KIN has a slightly weaker activity compared with CLV3 (Fig. 1d). Coinciding with the loss of the SAM, the expression of *WUS*, a downstream gene for CLV3 signalling[15,16], disappeared on the KIN treatment, indicating that KIN activates the CLV3 signalling pathway (Supplementary Fig. 2).

During the peptide treatment assays, we noticed that CLV3 and CLE25 reduce the thickening of vascular tissues in the hypocotyl. As shown in Fig. 1e, plants grown in liquid medium containing 10 μM CLV3 exhibited reduced radial growth of the stele. Contrary to expectations, KIN promoted the stele thickening, which is completely opposite to the effect of CLV3 (Fig. 1e). Such a stele-thickening activity was previously reported for CLE41/TDIF (refs 7,8) and indeed the CLE41-treated stele was thickened in our condition (Fig. 1e). Importantly, among the swapped peptides (3–16), only KIN showed the stele-thickening

activity similar to CLE41 (Fig. 1f), even though KIN was created by the swap of amino-acid residues between the two CLV3-type peptides, CLV3 and CLE25. We further analysed dose–response relationships in this assay (Fig. 1g). CLV3 exerted a negative effect on the stele growth at as low as 30 nM, whereas CLE41 showed a positive effect in higher concentrations (>1 μM). Interestingly, KIN exhibited negative effects in lower concentrations, whereas conversely it displayed positive effects in higher concentrations, demonstrating that KIN exerts both CLV3 and CLE41 types of activities by itself.

**Genetic dissection of bifunctional CLE bioactivity**. The dual activity of KIN could be attributed to its target receptors. *CLV1* and *CLV2* are receptor genes involved in CLV3 signalling and the SAMs of their loss-of-function mutants are resistant to CLV3 treatment[9,17,18]. Unlike the wild-type SAM (Fig. 1d), the mutant SAMs were maintained even after a 10-day treatment of 10 μM CLV3 and KIN (Fig. 2a), suggesting that KIN acts through intrinsic *CLV1/CLV2* pathways. Furthermore, *CLV2*, but not *CLV1*, is responsible for the root-shortening activity of CLV3 peptide[18]. Indeed, the *clv2-101* mutant was resistant to KIN and to CLV3 as well in terms of both the root length and RAM size (Supplementary Fig. 3a,b), indicating that KIN exerts the root-shortening activity via *CLV2*.

We next examined responses of these mutants to the peptides in the stele-thickening assay. As described above, a lower concentration (100 nM) of CLV3 or KIN reduced the stele width of wild-type plants, whereas a higher concentration (10 μM) of CLE41 or KIN thickened the stele (Figs 1g and 2b). In contrast to our assays where plants were exposed to CLV3 soon after germination, the CLV3-type inhibitory activity was not observed in the previous report in which 3-day-old seedlings were treated with the peptide[8], suggesting that sensitivity to CLV3 might differ by plant age. The response pattern of *clv1-101* mutant was similar to that of the wild type, suggesting that *CLV1* does not participate in CLE signalling in stele thickening. In contrast, *clv2-101* mutant was insensitive to the inhibitory activity of 100 nM CLV3 and KIN (Fig. 2b). Strikingly, 10 μM KIN showed a stronger effect than CLE41 in the *clv2-101* mutant (Fig. 2b). This phenomenon was similar to the previously reported synergistic effect of the simultaneous treatment of CLV3 and CLE41, which does not require functional *CLV2* (ref. 8). Simultaneous treatment of 10 μM CLV3 and 10 μM CLE41 showed a strong activity similar to the KIN treatment in *clv2-101*, although these activities were not observed in wild type in our experimental conditions (Fig. 2b). As *clv2-101* is insensitive to the inhibitory activity of CLV3 in stele thickening (Fig. 2b), this mutant serves as an ideal genetic background to detect the positive effect of CLV3-type peptides. Indeed, the dose–response assay in *clv2-101* showed that, in the presence of 10 μM CLE41, both CLV3 and KIN increase the stele thickening at concentrations above 1 μM (Supplementary Fig. 4). These data further support the notion that KIN exerts both activities of CLV3 and CLE41 by itself.

CLE41 treatment causes discontinued xylem strands in leaf vein due to its inhibitory activity on differentiation of undifferentiated vascular cells into xylem cells[7]. To further confirm whether KIN behaves similar to CLE41, we examined xylem strands after the KIN treatment. In this analysis, we used *clv2-101*, because the mutant is resistant to growth-inhibitory effects caused by CLV3 and KIN, and therefore we could obtain leaves at a comparable growth stage between different peptide treatments. We found that, similar to CLE41, the application of KIN caused inhibition of xylem differentiation,

which was not observed in CLV3 treatment (Fig. 2c). In summary, KIN possesses both CLV3- and CLE41-like activities in all assays examined (Fig. 2d).

To address whether KIN exerts the CLE41-like activity through the interaction with TDR, the only known receptor for CLE41, we performed peptide treatment experiments using *tdr-1* and *cle41-1*

mutants (Fig. 3a). Both of these mutants show reduction in stele width due to the loss of intrinsic CLE41-TDR signalling[8,19]. Application of CLE41 complemented the *cle41-1* mutant phenotype. As the KIN application also rescued the *cle41-1* mutant defect (Fig. 3a), KIN could function as CLE41. On the other hand, the receptor mutant *tdr-1* was insensitive to

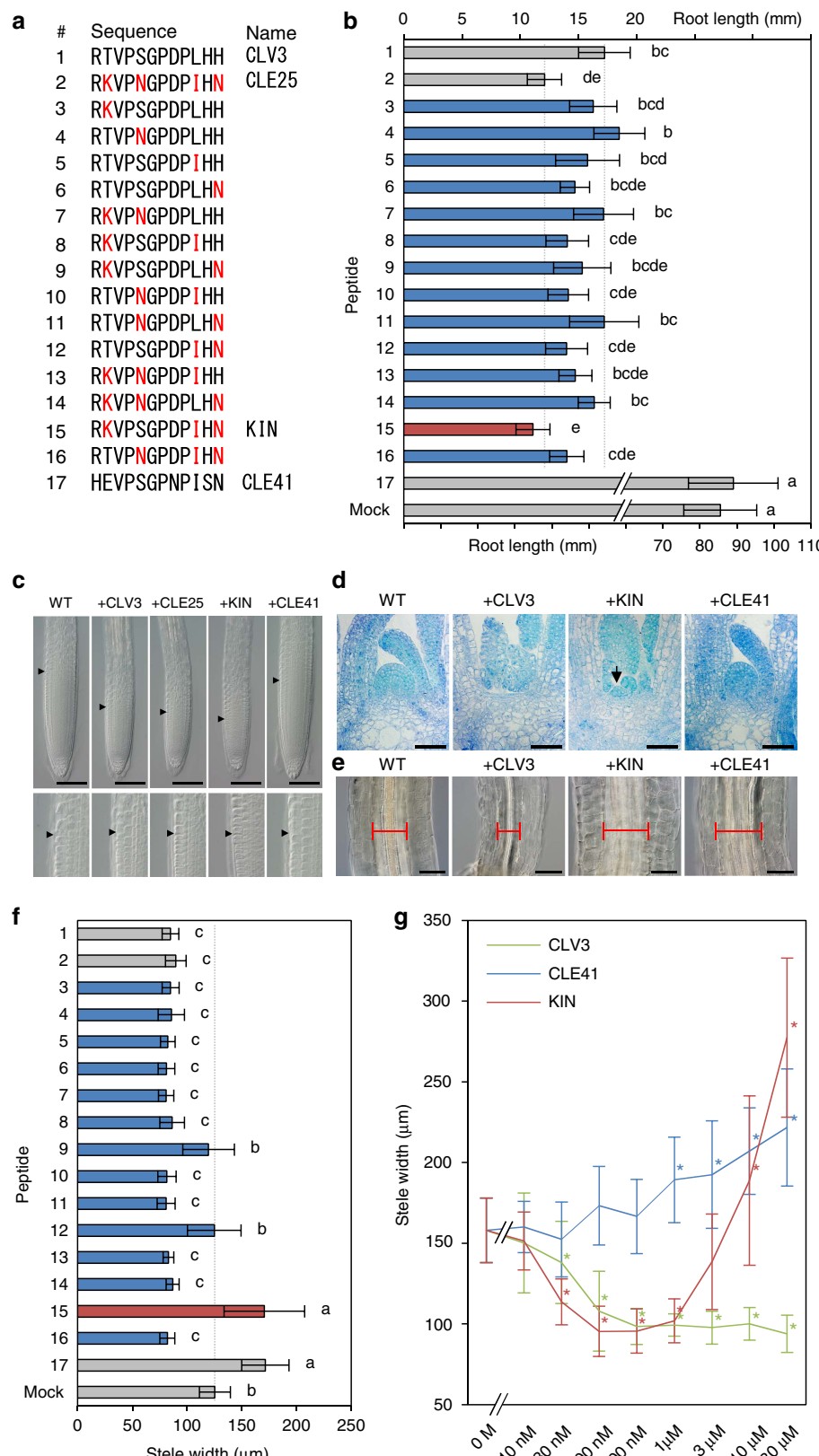

exogenous KIN and to CLE41 as well (Fig. 3a), suggesting that KIN acts through *TDR* to promote stele growth.

**Binding of bifunctional CLE peptide with CLV1 and TDR.** Our genetic analyses emphasize that the KIN peptide is capable of acting through both CLV1 and TDR. To examine the direct interaction between KIN and the receptors, we synthesized [$^{125}$I]-radiolabelled [(4-azidosalicyl)Lys$^2$]KIN (ASA-KIN) for photo-affinity labelling (Fig. 3b) and applied it to the membrane fractions containing receptor ectodomains expressed in tobacco cells. Although the bioactivity of ASA-KIN was reduced compared with the original KIN (Supplementary Fig. 5a–c), both of the CLV1 and TDR ectodomains were covalently labelled by [$^{125}$I]ASA-KIN (Fig. 3c). Under the same experimental condition, no interaction was detected between [$^{125}$I]ASA-KIN and the ectodomain of CLV2 (Supplementary Fig. 6), which is consistent with the previous study[20]. The interaction between the [$^{125}$I]ASA-KIN and CLV1 was competitively inhibited by excess amount of non-labelled CLV3 and KIN but not by CLE41, indicating that KIN specifically interacts with CLV1 at the CLV3-binding site. On the other hand, the binding of [$^{125}$I]ASA-KIN to TDR was competed by non-labelled CLE41 and KIN but not by CLV3, showing the specific binding of KIN to TDR at the CLE41-binding site. KIN exhibits a CLV3-type activity in a lower concentration, whereas it behaves as CLE41 in a higher concentration (Fig. 1g). This phenomenon might be caused by different binding manners of KIN to the two receptors, CLV1 and TDR. To address this possibility, we carried out competitive displacement of [$^{125}$I]ASA-KIN binding to the receptors with various concentrations of KIN according to the previously performed method[21]. [$^{125}$I]ASA-KIN binds to these receptors in a similar manner (Supplementary Fig. 7), suggesting that the binding of KIN to each receptor is not likely to be the major determinant of the difference in effective concentration for CLV3- and CLE41-type bioactivities. The difference is likely to be caused by other factors such as locations of target tissues and downstream signal transduction pathways.

**Role of specific residues for bioactivities of CLE peptides.** To elucidate the structural basis of the dual activity exerted by the hybrid peptide KIN, we examined the function of specific residues of CLE peptides. The amino-terminal residue of CLE peptides, which is conserved as R in CLV3-type peptides or H in CLE41-type peptides, has been recognized as an essential residue for their activities according to the previous Ala-scan assays[1,2,22]. Consistently, deletion of the N-terminal residue from KIN reduced its bioactivity at ~100-fold (Supplementary Fig. 8a–d, peptide 18). However, KIN exerts both CLV3 and CLE41 activities even though its N terminus is R, raising a possibility that the N terminus may not be important for the specificity of CLE activities. Indeed, KIN-H$^{1st}$ peptide also

showed a dual activity similar to KIN, both in stele-thickening and root-shortening assays (Supplementary Fig. 8a,b,e,f, peptide 19), indicating that the N-terminal residue is not responsible for the specificity. This was further supported by the fact that CLE41-R$^{1st}$ retained the CLE41 activity with no CLV3 activity (Supplementary Fig. 8a,b,e,f, peptide 20).

In addition to H$^{1st}$, CLE41 has the characteristic S$^{11th}$, which is conserved only among CLE41-type peptides in the CLE family[23]. We found that CLE41-H$^{11th}$ exhibited a dual activity, whereas CLE41-H$^{12th}$ showed only CLE41 activity (Supplementary Fig. 8a,b,e,f, peptides 21 and 22). CLE41-H$^{11th}$H$^{12th}$ showed CLV3 activity but lost CLE41 activity (Supplementary Fig. 8a,b,e,f, peptide 23). On the other hand, CLV3-S$^{11th}$ exhibited neither CLV3 nor CLE41 activity (Supplementary Fig. 8a,b,e,f, peptide 24). This H-to-S substitution also reduced the CLV3-type activity of KIN, although the effect was moderate (Supplementary Fig. 8a–d, peptide 25). Collectively, CLV3 requires H$^{11th}$ for its activity and the S$^{11th}$ of CLE41 prevents the peptide from displaying the CLV3 activity.

In the recently published crystal structures of the CLE41-TDR complex, the O$_\gamma$ atom of S$^{11th}$ forms a hydrogen bond with the ε-amino group of K$^{397th}$ of TDR[12,13]. We analysed the stability of the hydrogen bond at room temperature (300 K) in molecular dynamics (MD) simulations based on the atom coordinates of the CLE41-TDR complex[12]. We considered multiple alternative models for protonation states of titratable residues at 300 K, especially H$^{1st}$ of CLE41 and D$^{303rd}$ of TDR (Supplementary Information). The overall structure of CLE41 peptide was considerably more flexible at 300 K compared with the simulation at 77 K (mimicking the crystal), as shown by reduced fractions of native contacts (Supplementary Fig. 9a). The fraction of native contacts was lower in simulations with protonated D$^{303rd}$ of TDR (D$^{+303rd}$ versus D$^{303rd}$) at 300 K, while not influenced significantly by the protonation states of H$^{1st}$ of CLE41 (H$^{+1st}$ versus H$^{1st}$, Supplementary Fig. 9a). The higher flexibility was observed especially around the carboxy terminus of CLE41 as shown in Supplementary Fig. 9b by the root mean squared fluctuation of each C$_\alpha$-atom. Consequently, the duration of the hydrogen-bond formation between S$^{11th}$ and TDR was reduced at 300 K (45% and 16% of the entire simulation time with unprotonated and protonated D$^{303rd}$, respectively), compared with the stable hydrogen bond at 77 K (Supplementary Information). Collectively, it is likely that the interaction of S$^{11th}$ with TDR is significantly reduced at room temperature compared with the X-ray structure, which may explain why the mutation on S$^{11th}$ had little effect on the bioactivity of CLE41 in the previous report[1]. In contrast, N$^{12th}$, which is essential for the bioactivity[1], interacted with TDR >95% of the time (Supplementary Information), in spite of the increased flexibility at 300 K, which is due to the formation of a flexible network of hydrogen bonds with several residues of TDR.

The MD simulations raised a possibility that the side chain of the 11th residue of CLE peptides might not contribute

**Figure 1 | Identification of a bifunctional CLE peptide.** (**a**) Sequence alignment of CLE peptides. CLV3 (1), CLE25 (2) and CLE41 (17) are endogenously encoded sequences, whereas the others, including KIN (15), are intermediate sequences between CLV3 and CLE25. Residues changed from CLV3 to CLE25 are coloured red. (**b**) Effects of 1 μM peptides on 14-day-old root length. The upper scale is for treatment 1–16 and the lower scale is for treatment 17 and mock. The grey dashed lines indicate the levels for CLV3/CLE25 peptide treatment. (**c**) Effects of 1 μM peptides on 4-day-old RAM morphology. The arrowheads indicate the RAM areas. Lower panels show magnification of the boundary areas. (**d**) Effects of 10 μM peptides on 10-day-old SAM morphology. An arrowhead indicates a residual SAM. (**e**) Effects of 10 μM peptides on stele morphology in 10-day-old hypocotyls. Red bars indicate the stele width. (**f**) Effects of 10 μM peptides on 10-day-old hypocotyl stele width. (**g**) Dose–response relationships in 10-day-old stele width. Photos in **c–e** are representatives among three or more biologically independent samples. Data in **b,f** and **g** represent mean values ± s.d. (n = 13–20 in **b**, 13–16 in **f**, 12–16 in **g**, see Supplementary Data 1 for individual sample sizes). In **b,f**, means sharing the superscripts are not significantly different from each other in Tukey's HSD test, P < 0.05. Asterisks in **g** indicate a significant difference from mock treatment (0 M) in two-tailed Welch's t-test, P < 0.05. Scale bars, 100 μm (**c,e**) and 50 μm (**d**).

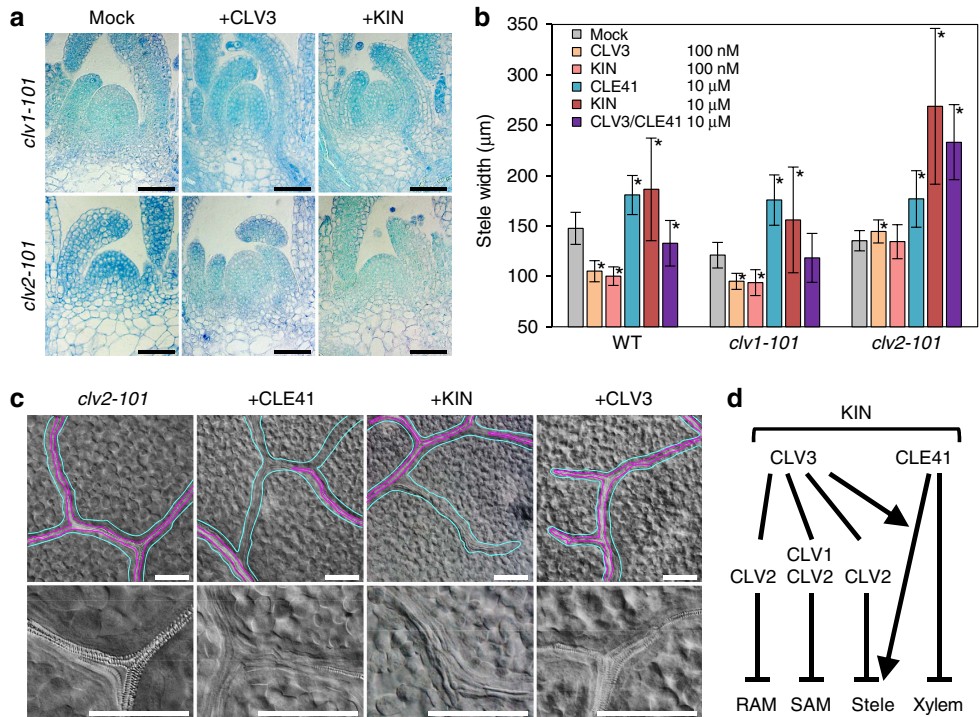

**Figure 2 | Genetic dissection of the KIN peptide activities.** (**a**) Effects of 10 μM peptides on 10-day-old mutant SAMs. Scale bars, 50 μm. (**b**) Effects of peptides on 10-day-old mutant stele width. The purple bars indicate simultaneous treatment of 10 μM CLV3 and 10 μM CLE41. Data represent mean values ± s.d. with asterisks, indicating a significant difference from mock treatment in each genetic background in two-tailed Weltch's *t*-test (*P* < 0.05, *n* = 12–16, see Supplementary Data 1 for individual sample sizes). (**c**) Effects of 10 μM peptides on 10-day-old *clv2-101* leaf veins. Cyan and violet lines indicate the vein and xylem strand, respectively, in the upper panels. It is noteworthy that both CLE41 and KIN treatment caused formation of veins without xylem strands. Lower panels are the magnification of veins. In CLE41- and KIN-treated plants, xylem strands with helical cell walls disappeared and narrow undifferentiated cells are observed. Scale bars, 200 μm. (**d**) Schematic representation of KIN bioactivity pathways. KIN exerts all the activities of CLV3 and CLE41 examined in this study, which are dependent or independent of CLV1 and CLV2 as indicated. Photos in **a** and **c** are representatives among three or more biologically independent samples.

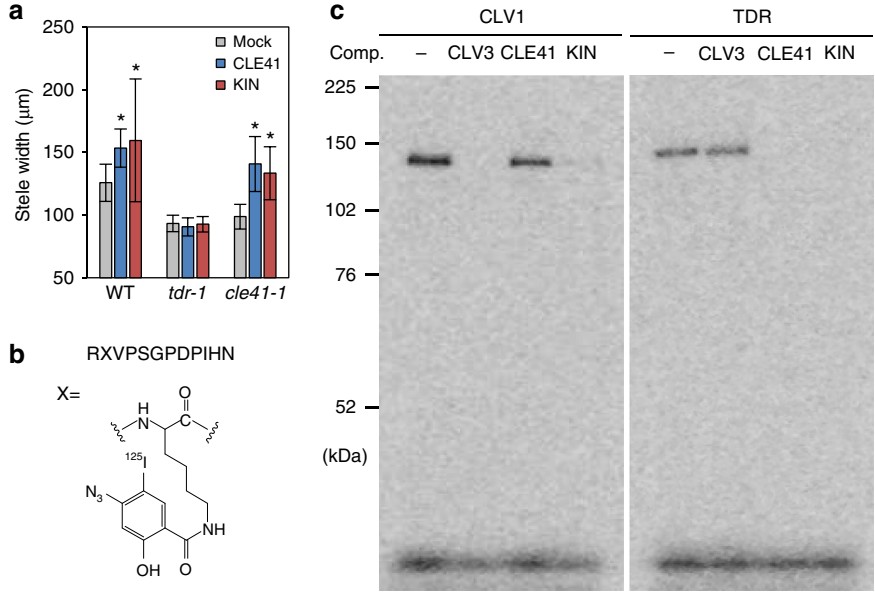

**Figure 3 | Direct interaction of the KIN peptide to CLV1 and TDR.** (**a**) Effects of 10 μM peptides on 10-day-old mutant stele width. Data represent mean values ± s.d., with asterisks indicating a significant difference from mock treatment in each genetic background in two-tailed Weltch's *t*-test (*P* < 0.05, *n* = 13–16, see Supplementary Data 1 for individual sample sizes). (**b**) Structure of the [125I] ASA-KIN photo-affinity probe. (**c**) Photo-affinity labelling of TDR and CLV1 ectodomains with [125I] ASA-KIN. Competition assays were performed with 1,000-fold excess amount of non-radio-labelled peptides as indicated above (comp.). Representative photos among two independent experiments are shown.

significantly to their affinities with their intrinsic receptors. To unequivocally address this, we examined the interaction between mutated peptides and receptors by the competitive displacement assay with [125I]ASA-KIN. As expected, CLV3-S[11th] retained the interaction with CLV1 (Supplementary Fig. 10, left), indicating that the H-to-S substitution does not compromise the binding of CLV3 with CLV1, even though it abolishes the bioactivity. Thus, the S[11th] of CLE peptides hampers CLV1 misactivation at a level other than the direct ligand–receptor interaction. Conversely, CLE41-H[11th], which is a bifunctional CLE peptide (Supplementary Fig. 8), interacted with both TDR and CLV1, although the interaction with CLV1 was not strong (Supplementary Fig. 10), showing the contribution of H[11th] to the interaction between CLE peptides and CLV1. The S[11th] of CLE41 is highly conserved in a number of flowering plants[23] and even in gymnosperms and ferns[24], although it is not required for CLE41 activity according to the previous Ala-scan assay[1], implying that unwanted dual activity, which may be detrimental to organized growth, has been selectively avoided during the molecular evolution of *CLE41* genes.

## Discussion

Here we demonstrate that the bifunctional CLE peptides, which have not been identified in nature, can be artificially engineered by using genetic variation among natural CLE peptides. Plant peptide hormones are typically encoded in a gene family, which contains small variations in the mature ligand sequences and each variation can have a unique role in exerting specific bioactivities. Some variations, such as S[11th] of CLE41, can be important to avoid unwanted cell signalling. In principle, sequence variations in natural peptide hormones are products under selection pressures in each evolutionary path. Importantly, peptides can take multiple mutational routes to reach or avoid specific bioactivities, as demonstrated in the engineering of bifunctional CLE peptides using different natural CLE peptides as starting materials. Thus, we propose that the hybrid synthesis of artificial peptides would provide a powerful methodology to use the natural genetic diversity as a source to mine cryptic bioactivities evolutionarily hidden in the genome and to engineer artificial cell signalling. For instance, given that genetic diversities in some peptide hormone families determine species-specific reproductive barriers[25,26], our approach could be used as a means to overcome reproductive barriers for the production of new beneficial plant/crop species.

## Methods

**Preparation of peptides.** Peptides were synthesized by Fmoc chemistry with a peptide synthesizer (CS136XT, CSBio). Hydroxyprolines were not included in the peptides used in this study. [125I]ASA-KIN was synthesized as described previously[10]. Fmoc-KIN (3.5 mg), 4-azidosalicylic acid succinimidyl ester (1.6 mg, Pearce) and NaHCO$_3$ (1.0 mg) were dissolved in 200 µl of 50% acetonitrile for 12 h in the dark with shaking at room temperature. Fmoc-ASA-KIN was purified by reverse-phase HPLC, lyophilized and deprotected in 25% piperidine in water for 1 h in the dark with gentle shaking at room temperature. The deprotected peptide was purified by reverse-phase HPLC to yield 1.8 mg of analytically pure ASA-KIN. ASA-KIN was further radioiodinated by the chloramine T method, as described previously[10]. The labelled peptide was purified by reverse-phase HPLC, to yield analytically pure [125I]ASA-KIN with specific radioactivity of 93 Ci mmol$^{-1}$.

**Photo-affinity labelling.** Aliquots (1,000 µg) of microsomal proteins for Halo-tagged receptors (CLV1-HT[10], TDR-HT[7] and CLV2-HT[20]) from tobacco BY-2 cells were suspended in 250 µl binding buffer (50 mM MES-KOH pH 5.5 with 100 mM sucrose) containing 30 nM [125I]ASA-KIN in the presence or absence of various concentrations of competitor peptides indicated in corresponding figure legends and then incubated for 10 min on ice. The bound and free [125I]ASA-KIN were separated by layering the reaction mixture onto 900 µl of wash buffer (50 mM MES-KOH pH 5.5 with 500 mM sucrose) and centrifuging for 5 min at 100,000 g at 4 °C. After discarding the supernatant, the pellet was irradiated on ice for 20 min

with an ultraviolet lamp (model ENF-260C/J (365 nm), Spectronics Co. Ltd) at a distance of 1 cm. The cross-linked membrane proteins were solubilized, immunoprecipitated by using HaloTag antibody and separated by SDS–PAGE on a 7.5% acrylamide gel. The dried gels were exposed to the bio-imaging plate (MS 2,025, Fujifilm) for 2 days at room temperature and the plates were analysed using a bio-imaging analyser (Typhoon FLA 900, GE).

**Plant materials.** Col-0 accession of *A. thaliana* was used as wild type. Loss-of-function mutants used in this study (*clv1-101*/WiscDsLox489-492B1, *clv2-101*/GK-686A09, *tdr-1*/SALK_002910 and *cle41-1*/CS92206) were described previously[7,17,18]. To express the β-glucuronidase (GUS) reporter gene under *WUS* promoter, 3.4 kb *WUS* promoter sequence was amplified with primers (5′-CAACGTCGACCACTCCTATGTTATTAGCTAAAATGTTTAG-3′ and 5′-CGGGATCCGTGTGTTTGATTCGACTTTTGTTC-3′), and ligated into SalI–BamHI restriction sites of the binary vector pBI101.1. Col-0 plants were transformed with *Agrobacterium tumefaciens* (GV3101 Mp90) using the floral dip method[27].

**Bioassay.** For root-length measurement, plants were germinated and grown vertically on half-strength Murashige and Skoog (MS) medium supplemented with 1% sucrose and peptide/control solution at 22 °C under continuous light. To observe the RAM, 4-day-old roots were excised and mounted in clearing solution (chloral hydrate/glycerol/water = 8:1:2) before imaging with light microscope (Axio Imager.A2, Zeiss).

To observe the SAM, plants were germinated and grown at 22 °C under continuous light on half-strength MS medium supplemented with 1% sucrose and 10 µM peptide/control solution. To make sections, roots and leaves were cut off from 10-day-old seedlings, then fixed in FAA solution (50% ethanol:10% formalin:5% acetic acid in water) and embedded into Technovit 7,100 resin according to the manufacturer's instructions (Heraeus Kulzer). Four-micrometre-thin sections were made using a microtome (RM2235, Leica), stained with 0.05% toluidine blue and mounted in Entellan New (Merck) before observation with a light microscope (Axio Imager.A2, Zeiss).

For the observation of stele and leaf vein, seeds were germinated and cultured with shaking at 110 r.p.m. at 22 °C under continuous light in liquid half-strength MS medium supplemented with 1% sucrose and peptide/control solution. 10-day-old seedlings were fixed in a 1:3 mixture of acetic acid/ethanol, washed with water and mounted in clearing solution (chloral hydrate/glycerol/water = 8:1:2) before imaging with a light microscope (Axio Imager.A2, Zeiss).

**GUS staining.** Plants were fixed in 90% acetone at −20 °C overnight, washed twice with 100 mM sodium phosphate buffer (pH 7.2), and then incubated in X-gluc solution (1 µM 5-bromo-4-chloro-3-indolyl-β-D-glucuronic acid (Wako), 100 mM sodium phosphate pH 7.2, 10 mM EDTA, 0.1% Triton X, 10 mM potassium ferrocyanide and 10 mM potassium ferricyanide) for 2 h at 37 °C. The GUS-stained samples were cleared with 70% ethanol and mounted with clearing solution (chloral hydrate:glycerol:water, 8:1:2) before imaging with light microscope (Axio Imager.A2, Zeiss).

**Statistical analysis.** Statistical analysis was performed with Excel (Microsoft) or R (www.R-project.org). Two-sided Welch's *t*-test was performed with Excel. For the multiple comparison, analysis of variance and Tukey's honest significant difference test were performed with R-package 'agricolae'. The sample size was determined based on the previous studies[7,13]. Exact sample size for each data is shown in Supplementary Data 1.

**MD simulation.** Detailed methods of structure preparation for simulation and calculation of p$K_a$ of titratable residues are provided in Supplementary Note 1.

**Data availability.** The authors declare that all data supporting the findings of this study are available within the manuscript and its Supplementary Information files or are available from the corresponding authors upon request.

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

## Acknowledgements

We thank Hiroo Fukuda for *tdr-1* and *cle41-1* lines, the ABRC Stock Center for *clv1-101* and *clv2-101* lines, Radioisotope Research Center of Nagoya University for instrumental support in radioisotope experiments, Peptide/Protein Center of WPI-ITbM in Nagoya University for support in peptide synthesis and Toshiaki Tameshige for valuable comments. This research was supported by MEXT/JSPS KAKENHI (grant numbers JP26113507, JP26291057 and JP16H01237 to K.U.T; JP25114511, JP26113707 and JP16H01462 to N.U.; JP25221105 and JP15H05957 to Y.M; JP26113520, JP16H01234 and JP25840111 to H.S.) and Howard Hughes Medical Institute (HHMI) and Gordon and Betty Moore Foundation (GBMF) (grant number GBMF3035 to K.U.T). K.U.T. is an HHMI-GBMF Investigator. Y.H. and K.W. are JSPS Postdoctoral Fellows. Y.H. was an HFSP Postdoctoral Fellow.

## Author contributions

Y.H. conceived the research. Y.H. jointly designed the research with N.U. with the assistance of all co-authors. Y.H. performed bio-assays. H.S. performed the photo-affinity labelling experiments. K.W. performed MD simulations. N.U. constructed and established *WUSprom:GUS* transgenic line. H.S. and Y.H. synthesized the photo-affinity probes. H.S. and Y.M. provided the receptor membrane fractions. All authors analysed data. Y.H. wrote the manuscript with N.U and K.U.T, and with inputs from K.W. and H.S.

## Additional information

**Competing financial interests:** The authors declare no competing financial interests.

**Publisher's note**: 

