## [Peer Review File · Nature Communications]

Reviewers' comments:

Reviewer #1 (Remarks to the Author):

A. Summary of the key results

The manuscript by Hirakawa et al. briefly but very elegantly describes a novel strategy to capture the mechanisms underlying peptide and receptor function in plants. By creating hybrid peptides based on the sequences of two peptides (CLV3 and CLE25) they were able to create a peptide (KIN) that was able to bind to the receptors for CLV3 - CLV1 - and CLE25 - TDR, respectively. They use this peptide to dissect CLV1 and CLV2 dependent and independent signaling pathways which might be covered by the synthetic KIN peptide and the overlap to CLE41 signaling. Moreover, in addition to providing novel insights into receptor function, the work also raises new and rather fundamental questions about receptor/ligand interactions in particular with respect to the relevance of individual amino acid residues for binding.

B. Originality and interest: if not novel, please give references

In this manuscript the authors have used a creative and original approach to dissect peptide function. The work is very interesting to plant science but moreover to any scientist studying protein-protein interactions (not limited to peptide ligands and receptors) as it raises fundamental questions about signaling specificity versus flexibility and the evolution of specific signaling modules.

C. Data & methodology: validity of approach, quality of data, quality of presentation

The authors have carefully designed the research strategy and conducted the work. They justify their approach and provide some context to their work. All data has been very carefully described and mostly been very well presented (see some suggestions under F.)

D. Appropriate use of statistics and treatment of uncertainties

All data has been statistically analyzed. Statistical tests and treatments are adequate for the presented data.

E. Conclusions: robustness, validity, reliability

The conclusions presented by the authors are justified and overall well presented. They could however add a short paragraph describing, even in a speculative manner, how future structural and evolutionary work could expand on the work presented in this manuscript.

F. Suggested improvements: experiments, data for possible revision

The work in this manuscript is very well designed and carried out and in essence stands as is. However, I have few small suggestions that might improve the final article.

- Figure 1b: The authors use a dose response curve to show that the function of KIN peptide in control of stele width is dependent on the concentration of the peptide. A similar dose response for CLV3, CLE41 and KIN on root length could be appropriate to show. I find this particularly interesting since the peptide concentration used in figure 1b is 1 μM while KIN peptide reduced stele width at 1 μM but led to a wider stele at concentrations of 10 and

30 μ M.

- Figure 1c: The images in the figure are very small and the differences described by the authors (indicated by arrowheads) are rather hard to see (impossible in a printed version of the manuscript). If the authors have available images of roots stained with propidium iodine (for example as described here: <https://sites.duke.edu/benfey/protocols/confocal-imaging-of-roots/>) it could drastically enhance the figure and may look more obvious to the reader. Alternatively, increasing image size and providing a zoomed image of the region highlighted with arrowheads will be helpful. The same applies to supplementary figure 2b.
- Figure 1f: very minor comment, the bar graph might be increased in size to match panel 1b.
- Figure 2b: It seems that 10 μ M KIN and CLV3/CLE41 peptides have a stronger effect on stele width in *clv2* mutant background compared to wild type plants. This could be mentioned in the text together with a short speculation on the reason.
- Figure 2c: The legend does not describe the amount of peptides used in the experiment (which is critical given the results in figures 1g and 2b). Also, in the light of the results in figure 2b, if data on the effect of the peptides on leaf veins is available using different peptide concentrations this should be included as supplementary information.
- Figure 3c: Does KIN peptide bind to CLV1 and TDR with the same affinity? While not absolutely required, analyzing the binding affinity to both receptors could be very interesting for discussion given the intriguing behavior of KIN in the regulation of stele width (Figure 1g and 2b).
- Supplementary figure 4: The authors describe that altering the N-terminal amino acid in KIN from R to H did not affect its function even though an N-terminal R in CLV3-type peptides has been described as important for activity. This assay makes me wonder if a peptide where the N-terminal amino acid (R or H) is deleted would still be functional. Similarly, I was left to wonder about the activity and function of a KIN peptide with a serine residue in position 11. I realize that this is beyond the scope of the current manuscript but should data be available it would be great as an addition to the supplementary information.

G. References: appropriate credit to previous work?

References are adequate.

H. Clarity and context: lucidity of abstract/summary, appropriateness of abstract, introduction and conclusions

The text in the manuscript is very clear and pleasant to read. The abstract and introduction are appropriate. In the conclusions the authors might include a more thorough outlook on how future structural and evolutionary studies on the basis on the described work could lead to very exciting novel insights into peptide/receptor signaling.

Reviewer #2 (Remarks to the Author):

Comments to the authors:

The presented study "Cryptic bioactivity capacitated by synthetic hybrid plant peptides" by Hirakawa et al. reports a series of synthetic CLV3/CLE25 hybrid peptides. Within this screen the authors uncover a peptide variant termed KIN which, when applied to plants, triggers phenotypes resembling both CLV3 (shorter root growth, smaller RAM and SAM as well as reduced root stele width) and CLE41 (bigger stele width and xylem development in leaf veins) which is surprising as the actions of CLV3 and CLE41 have been reported to not overlap. The leucine rich repeat receptor kinases CLV1 and CLV2 are involved in CLV3 signaling and the authors report that *clv2* mutant plants are resistant to KIN application for all phenotypes while *clv1* is resistant to KIN application for the SAM phenotype but sensitive for all the other phenotypes shown. Additionally, the null mutant of the CLE41 receptor, TDR, is insensitive to KIN application while KIN application rescues the *cle41* root stele morphology phenotype. Photo-affinity labeling shows the binding of KIN with both CLV1 and TDR which can be outcompeted by the native ligand, CLV3 and CLE41, respectively. Finally, by further alternating the KIN and CLE41 peptide sequence the authors suggest an important role of the c-terminus for the signaling specificity of CLE peptides. I have a few comments which I summarize below:

1. Do the peptides used in this study include a hydroxyproline?
2. Please explain the meaning of the letters in Fig. 2B.
3. The authors report a dose-dependent effect of the KIN peptide on stele width but no interpretation or discussion of the phenomenon is given. Could it be that the peptide binds to different receptors with different affinities? Along these lines, does addition of 3-10 μ M CLE41 to 100 nM KIN increase the stele width?
4. Whitford et al. PNAS 2008 show that CLE6p (an A-type CLE peptide like CLV3 which has similar effects on the RAM) application does not change stele width. Could the authors elaborate on this apparent difference?
5. To avoid confusion about CLV2, the authors should state the CLV2 does not directly bind CLV3 (Shinohara and Matsubayashi, Plant Journal 2013).
6. Whitford et al., PNAS 2008 report that both *clv1* and *clv2* mutant plants show WT-responses in stele morphology when CLE6 and CLE41 are applied but this is not discussed in the text.
7. The *clv2* but not the *clv1* mutant plants were hyposensitive to KIN application for all tested phenotypes. The only phenotype in which *clv1* differed from WT plants was in SAM development. Therefore, it should be addressed if KIN can bind to CLV2. Also the potential role of CLV2 in TDR-mediated signaling is not mentioned or discussed. *tdr-1* plants are insensitive to KIN and CLE41 induced stele enlargement but the *clv2-101* mutant appears to be hypersensitive to only KIN but not to CLE41 treatment. Please discuss.
8. What were the concentrations of the peptides used for the assays shown in figure S4?
9. To validate the hypothesis put forth in the discussion the authors could test TDR K397M and CLV1 M389K mutants in the photo-affinity labeling assay which would help to understand the specificity of CLE peptide binding to LRR-RKs

Reviewer #3 (Remarks to the Author):

This manuscript contains some interesting observations on the behaviour of several peptides related to CLV3. The authors initially looked at the difference between the activities of the CLV3 and CLE25 two peptides that have broadly similar spectrum of activity albeit with different activities on particular phenotypes such as root shortening. They were able to show that a particular synthetic peptide, that was essentially a hybrid between CLV3 and CLE25, also exhibited activity in similar manner to the TDIF peptide of CLE41 that normally exhibits a quite different, and in some cases opposite, set of activities to CLV3 and CLE25. It certainly is an interesting observation

The manuscript is clearly written and the conclusions sound. There appears to be no obvious problems with the data. Some of the error bars are quite large, but the appropriate statistical test appear to have been carried out. The respective activities of the hybrid "KIN" peptides requires the appropriate receptor kinase for either CLE41 or CLV3- like activity. I am slightly unclear about what is the central message of this paper. The authors show that they can generate what is essentially a novel peptide with novel bifunctional activity that is representative of 2 classes of CLE peptides. The claim is that this could be the route to entirely new classes of peptides with novel activities that could be used for different applications. It also sheds some light on the structure function relationship of CLE peptides. Close inspection of CLV3 CLE25 and CLE41 show that in fact there is a lot of similarity between CLV and CLE41 the central VPSGPDP is common to both that suggest that the 'class specificity lies outside this region. Of the 3 remaining amino acids at the N-terminus of CLE41 two are shared with CLE25. So the best hybrid of CLV3/CLE25 would have 9 of 10 identical amino acids at the N-terminus. Peptides, such as no. 12, that contain this sequence appear to have both kinds of activities. Incorporating K at position 2 to generate KIN enhances bifunctional activity. While this is an interesting observation it is unclear whether this is ever likely to be generally applicable as a useful means of generating novel peptides activities. There is no indication that this method might be somehow be applied more generally. Is this just a one off that results from the particular sequence similarities between CLE25/CLV3 and CLE41? In which case is it unlikely ever to be more useful than just controlling the expression of 2 peptides independently.

From the point of view about what this tells us about the structure function relationship between CLE peptides this appears to be a great starting point that raises some interesting questions, however few of these questions have been addressed in the manuscript and in this sense it is rather premature. It is also unclear whether the cryptic activity reveal has any biological meaning and is important as part of the CLE peptide signalling network.

Point-by-point Responses to Reviewers (in bold)

Changes are tracked in yellow in the revised manuscript.

To address the comments by reviewers, we have performed additional experiments. 6 new figures, designated as Supplementary Figures 1, 4, 6, 7, 9 and 10, are included in the revised manuscript.

Reviewer #1:

A. Summary of the key results

The manuscript by Hirakawa et al. briefly but very elegantly describes a novel strategy to capture the mechanisms underlying peptide and receptor function in plants. By creating hybrid peptides based on the sequences of two peptides (CLV3 and CLE25) they were able to create a peptide (KIN) that was able to bind to the receptors for CLV3 - CLV1 - and CLE25 - TDR, respectively. They use this peptide to dissect CLV1 and CLV2 dependent and independent signaling pathways which might be covered by the synthetic KIN peptide and the overlap to CLE41 signaling. Moreover, in addition to providing novel insights into receptor function, the work also raises new and rather fundamental questions about receptor/ligand interactions in particular with respect to the relevance of individual amino acid residues for binding.

B. Originality and interest: if not novel, please give references

In this manuscript the authors have used a creative and original approach to dissect peptide function. The work is very interesting to plant science but moreover to any scientist studying protein-protein interactions (not limited to peptide ligands and receptors) as it raises fundamental questions about signaling specificity versus flexibility and the evolution of specific signaling modules.

C. Data & methodology: validity of approach, quality of data, quality of presentation

The authors have carefully designed the research strategy and conducted the work. They justify their approach and provide some context to their work. All data has been very carefully described and mostly been very well presented (see some suggestions under F.)

D. Appropriate use of statistics and treatment of uncertainties

All data has been statistically analyzed. Statistical tests and treatments are adequate for the presented data.

E. Conclusions: robustness, validity, reliability

The conclusions presented by the authors are justified and overall well presented.

We appreciate Reviewer 1 for acknowledging the importance and novelty of our findings.

They could however add a short paragraph describing, even in a speculative manner, how future structural and evolutionary work could expand on the work presented in this manuscript.

A1. Reviewer 3 also gave a comment related to this suggestion. As suggested, we added sentences in the closing paragraph to briefly describe how our findings will provide insights into future work (p11-12, Line 245-262). The points are the following. Firstly, we have engineered bifunctional CLE peptides, which has never been identified in nature, suggesting that evolutionarily hidden cryptic bioactivities can be mined by our hybrid synthesis approach. We have engineered two distinct bifunctional CLE peptides from different endogenous CLE peptides as starting materials, indicating that there can be multiple mutational routes to reach or avoid specific bioactivities as seen in evolution. Collectively, we believe that the hybrid synthesis will provide a powerful tool for engineering artificial cell signaling, which may include agricultural benefits such as overcoming reproductive barriers.

F. Suggested improvements: experiments, data for possible revision

The work in this manuscript is very well designed and carried out and in essence stands as is. However, I have few small suggestions that might improve the final article.

- Figure 1b: The authors use a dose response curve to show that the function of KIN peptide in control of stele width is dependent on the concentration of the peptide. A similar dose response for CLV3, CLE41 and KIN on root length could be appropriate to show. I find this particularly interesting since the peptide concentration used in figure 1b is 1 μ M while KIN peptide reduced stele width at 1 μ M but led to a wider stele at concentrations of 10 and 30 μ M.

A2. As suggested, we examined dose responses of CLV3, CLE25, CLE41 and KIN (Supplementary Fig 1). The results show the effective concentration of peptides in root shortening assay is lower than those in other phenotypes. We will further discuss this point later (see A5 later), combined with other phenotypes and the KIN-receptor affinities. In the root shortening assay, we also found that CLV3 starts to show a bioactivity (~10nM) at a lower concentration than CLE25 (~100nM). These data provide another example that highlights the importance of minor variations in peptide sequences for fine-tuning bioactivities. In order to focus on the main finding (bifunctional CLE peptide) of this manuscript, we described the result briefly (p3 Line 58-61) rather than emphasizing this point.

- Figure 1c: The images in the figure are very small and the differences described by the authors (indicated by arrowheads) are rather hard to see (impossible in a printed version of the manuscript). If the authors have available images of roots stained with propidium iodine (for example as described here: <https://sites.duke.edu/benfey/protocols/confocal-imaging-of-roots/>) it could drastically enhance the figure and may look more obvious to the reader. Alternatively, increasing image size and providing a zoomed image of the region highlighted with arrowheads will be helpful. The same applies to supplementary figure 2b.

A3. We appreciate the suggestion. We provided enlarged photos in Figure 1c and Supplementary Figure 3b.

- Figure 1f: very minor comment, the bar graph might be increased in size to match panel 1b.

A4. We fixed the size of the panels accordingly (Figure 1).

- Figure 2b: It seems that 10 μ M KIN and CLV3/CLE41 peptides have a stronger effect on stele width in *clv2* mutant background compared to wild type plants. This could be mentioned in the text together with a short speculation on the reason.

A5. The *clv2* mutant is insensitive to the negative effect of CLV3-type peptides in stele thickening (Fig. 2b). Therefore, *clv2* mutant senses only growth-promoting effect of CLV3/CLE41 and KIN, and thus provides a sensitized background for the positive effects. We described this point in p6 Line 118-120. Also in A.12 (see later), we used this ideal, sensitized background to examine the positive effect of CLV3-type activity by a dose responses analysis (p6 Line 120-123, Supplementary Fig 4).

- Figure 2c: The legend does not describe the amount of peptides used in the experiment (which is critical given the results in figures 1g and 2b). Also, in the light of the results in figure 2b, if data on the effect of the peptides on leaf veins is available using different peptide concentrations this should be included as supplementary information.

A6. We added the information of the concentration used in the legends. As the effect on leaf veins was difficult to measure quantitatively compared with the other phenotypes examined in this study, we did not carry out assays using different peptide concentrations.

• Figure 3c: Does KIN peptide bind to CLV1 and TDR with the same affinity? While not absolutely required, analyzing the binding affinity to both receptors could be very interesting for discussion given the intriguing behavior of KIN in the regulation of stele width (Figure 1g and 2b).

A7. ³H-labelled peptides are needed to measure the exact binding affinities. Unfortunately, however, we could not obtain them due to a technical problem. Instead of examining the affinity itself, we examined the amounts of KIN peptides needed to competitively displace the ASA-KIN probe bound to CLV1 and TDR receptors in order to address whether the probe binds to the two receptors in a similar or different manner. This strategy was used previously to compare the affinities of two peptides to one receptor (Okamoto et al. 2013 *Nat. Commun.* 4, 2191). Our results show that ASA-KIN binds to the two receptors in a similar manner, suggesting that other factors, such as locations of target tissues and/or downstream signal transduction pathways, may cause the differences in effective concentration for CLV3- and CLE41-type bioactivities conferred by the KIN peptide. We described this in p7-8 Line 161-172 of the revision.

This interpretation is also consistent with previous reports. For instance, CLE41 peptide is active in much lower concentrations in cell-culture assay (as low as 30 pM, Ito et al., 2006 *Science* 313, 842-845) than in the seedlings, which may be due to the difference in permeability of peptides into target cells/tissues between *in vitro* and *in vivo*. We also showed that, in stele thickening, CLV3-type peptides need different concentrations to exert growth inhibitory (30 nM) or promoting (> 1 μM) activities (Supplementary Fig. 4).

• Supplementary figure 4: The authors describe that altering the N-terminal amino acid in KIN from R to H did not affect its function even though an N-terminal R in CLV3-type peptides has been described as important for activity. This assay makes me wonder if a peptide where the N-terminal amino acid (R or H) is deleted would still be functional. Similarly, I was left to wonder about the activity and function of a KIN peptide with a serine residue in position 11. I realize that this is beyond the scope of the current manuscript but should data be available it would be great as an addition to the supplementary information.

A8, To address these points, we performed bioassays using the N-terminal deletion of KIN (KIN-1Δ) and H11S substitution of KIN (KIN-11S). The results are shown in Supplementary Fig. 8c (stele) and 8d (root), respectively. KIN-1Δ (Supplementary Fig. 8c, peptide 18) showed about 100-fold reduction in both root and stele activities (p8, Line 180-181) as expected. Interestingly, the activity of CLV3-11S (Supplementary Fig. 8e, f,

peptide 24), is even weaker than KIN-15. Consistently, KIN-11S (Supplementary Fig. 8c, d, peptide 25), also reduced its CLV3-type activity, but unexpectedly the effect was moderate (p9, Line 195-197). These data provide yet another example illustrating the importance of minor variations in peptide sequences for fine-tuning bioactivities.

G. References: appropriate credit to previous work?

References are adequate.

H. Clarity and context: lucidity of abstract/summary, appropriateness of abstract, introduction and conclusions

The text in the manuscript is very clear and pleasant to read. The abstract and introduction are appropriate. In the conclusions the authors might include a more thorough outlook on how future structural and evolutionary studies on the basis on the described work could lead to very exciting novel insights into peptide/receptor signaling.

We thank Reviewer 1 for positive and enthusiastic comments.

Reviewer #2 (Remarks to the Author):

Comments to the authors:

The presented study "Cryptic bioactivity capacitated by synthetic hybrid plant peptides" by Hirakawa et al. reports a series of synthetic CLV3/CLE25 hybrid peptides. Within this screen the authors uncover a peptide variant termed KIN which, when applied to plants, triggers phenotypes resembling both CLV3 (shorter root growth, smaller RAM and SAM as well as reduced root stele width) and CLE41 (bigger stele width and xylem development in leaf veins) which is surprising as the actions of CLV3 and CLE41 have been reported to not overlap. The leucine rich repeat receptor kinases CLV1 and CLV2 are involved in CLV3 signaling and the authors report that *clv2* mutant plants are resistant to KIN application for all phenotypes while *clv1* is resistant to KIN application for the SAM phenotype but sensitive for all the other phenotypes shown. Additionally, the null mutant of the CLE41 receptor, TDR, is insensitive to KIN application while KIN application rescues the *cle41* root stele morphology phenotype. Photo-affinity labeling shows the binding of KIN with both CLV1 and TDR which can be outcompeted by the native ligand, CLV3 and CLE41, respectively. Finally, by further alternating the KIN and CLE41 peptide sequence the authors suggest an important role of the c-terminus for the signaling specificity of CLE peptides. I have a few comments which I summarize below:

1. Do the peptides used in this study include a hydroxyproline?

A9. We did not use hydroxyprolines throughout this manuscript. Their contribution to the bioactivity has been shown to be rather minor (Ito et al., 2006 *Science* 313, 842-845; Kondo et al. 2006 *Science* 313, 845-848) unless they are further glycosylated (Ohyama et al. 2009 *Nat. Chem. Biol.* 5, 578-580). We mentioned this point in the materials and methods (p13, Line 291-292).

2. Please explain the meaning of the letters in Fig. 2B.

A10. In Fig. 2B, the letters indicate the peptides we applied to plants. In particular, "CLV3/CLE41" means simultaneous treatment of 10 μ M CLV3 and 10 μ M CLE41. To clarify these points, we added the information both in the Fig.2b legend and the main text (p6, Line 115-118).

Perhaps "Fig. 2B" in this comment might refer to "Fig. 1B". In this case, the letters in Fig.1B show the result of Tukey's HSD test to examine statistical differences among effects of the treated peptides. The treatments sharing the same letters are not significantly different from each other in Tukey's HSD test. This explanation is described in the figure legend.

3. The authors report a dose-dependent effect of the KIN peptide on stele width but no interpretation or discussion of the phenomenon is given. Could it be that the peptide binds to different receptors with different affinities? Along these lines, does addition of 3-10 μ M CLE41 to 100 nM KIN increase the stele width?

A11. As we described in A5 to respond to the reviewer 1's comment, CLV3 and KIN exert a negative effect on stele thickening via CLV2 (Fig. 2b). This negative effect was observed even in a low dosage of CLV3 or KIN as low as 30 nM (Fig. 1g). On the other hand, it has been reported that CLV3-type peptides also exert a positive effect in a higher concentration when simultaneously applied with CLE41 (Whitford et al. 2008 *PNAS* 105, 18625-18630). CLE41 requires TDR for this positive effect, while the receptor for CLV3-type peptides is still unknown. Therefore, it is likely that KIN exerts a positive effect via both TDR and the yet unknown receptor. These pathways are summarized in Fig 2d.

We agree with Reviewer 2 that it is important to clarify the effective

concentration for each CLE bioactivity. However, the “positive” effect of CLV3 can be observed only when simultaneously treated with CLE41 (Fig 2; Whitford et al. 2008 *PNAS* 105, 18625-18630). Therefore, we additionally examined a dose-dependent effect of the CLV3’s “positive” effect by using *clv2-101* mutant, which is an ideal background for this analysis due to its insensitivity to the “negative” effect of CLV3. We fixed the CLE41 concentration at 10 μ M for these experiments (Supplementary Figure 4 in our revision). Our results clearly show that >1 μ M of both CLV3 and KIN peptides are needed to increase the stele width. We described these results in p6 Line 118-123 in the revision.

4. Whitford et al. *PNAS* 2008 show that CLE6p (an A-type CLE peptide like CLV3 which has similar effects on the RAM) application does not change stele width. Could the authors elaborate on this apparent difference?

A12. The conditions used in the two studies are slightly different. In particular, Whitford et al. used 3-day-old plants germinated on agar plates for peptide treatment. On the other hand, in our study, plants were exposed to peptides soon after germination, and thus the age of plants may affect their sensitivities to CLV3. We mentioned this in p5 Line 104-108.

5. To avoid confusion about CLV2, the authors should state the CLV2 does not directly bind CLV3 (Shinohara and Matsubayashi, *Plant Journal* 2013).

A13. We added the suggested statement (p8 Line 153-156), accompanied by the biochemical assay which demonstrate that CLV2 does not directly bind KIN peptide, either. The data is now included in our revision (please see A15 for details).

6. Whitford et al., *PNAS* 2008 report that both *clv1* and *clv2* mutant plants show WT-responses in stele morphology when CLE6 and CLE41 are applied but this is not discussed in the text.

A14. Thank you for catching this. We corrected the description accordingly in p5-6 Line 114-115.

7. The *clv2* but not the *clv1* mutant plants were hyposensitive to KIN application for all tested phenotypes. The only phenotype in which *clv1* differed from WT plants was in SAM development. Therefore, it should be addressed if KIN can bind to CLV2. Also the potential role of CLV2 in TDR-mediated signaling is not mentioned or discussed. *tdr-1* plants are insensitive to KIN and CLE41 induced stele enlargement but the *clv2-101* mutant appears to be hypersensitive to only

KIN but not to CLE41 treatment. Please discuss.

A15. As we described in A13, no direct interaction was detected between CLV2 and KIN (p7 Line 153-156, Supplementary Fig 6), which is consistent with the previous report showing no interaction between CLV3 and CLV2 (Shinohara and Matsubayashi. 2015 *Plant J.* 82, 328-336).

Our discussion on the reason why the *clv2* mutant is hypersensitive to KIN compared to CLE41 is the following (please also see Fig.2d). As we described in A11, *clv2-101* mutant is insensitive to the negative effect of CLV3, and thus sensitized to its positive effect. Importantly, CLE41 alone exerts a positive effect via TDR signaling, and the positive effect of CLV3 can be seen only when treated simultaneously with CLE41. Because KIN alone can activate both pathways, it triggers the synergistic effect by itself (as if CLV3 and CLE41 are co-treated). In particular, since *clv2* is sensitized to the positive effect, *clv2* is hypersensitive to KIN. In contrast, CLE41 activates TDR signaling with no synergistic effect. Therefore, the *clv2-101* mutant appears to be hypersensitive to only KIN but not to CLE41 treatment. We explained this point in p6 Line 117-121.

8. What were the concentrations of the peptides used for the assays shown in figure S4?

A16. We added the information on concentrations in the legend of Supplementary Fig.8 e,f (stele 10 μM / root 1 μM). They are same as those in Fig 1b and 1f, respectively. Thanks for catching this.

9. To validate the hypothesis put forth in the discussion the authors could test TDR K397M and CLV1 M389K mutants in the photo-affinity labeling assay which would help to understand the specificity of CLE peptide binding to LRR-RKs

A17. Since we did not have adequate time for in vivo receptor mutagenesis in this revision, we instead performed mutation analyses, computationally and experimentally, on the peptide side. We believe that the binding assays of mutant peptides are complementary to the suggested experiments and will essentially address the same question.

Firstly, we conducted Molecular Dynamics (MD) simulations based on the crystal structure of the TDIF-TDR ligand-receptor complex recently reported by Zhang et al. 2016 *Cell Res.* 26, 543-555. The details are shown in p9-10 Line 199-223 and Supplementary Fig. 9. These MD simulations have been performed by two theoretical

chemists, Dr. Kai Welke and Prof. Stephan Irle, who are now included as co-authors. The simulations predict that the peptide is more flexible in room temperature (300K) compared to the crystal structure determined at a very cold temperature (77K), and the interaction between the side chain of S^{11th} of CLE41 and K^{397th} of TDR is not stable at the room temperature. These results from the MD simulations raised a possibility that the side chain of the 11th residue of CLE peptides might not contribute significantly to their affinities with their intrinsic receptors. Therefore, we next examined this possibility by biochemical experiments. We analyzed the interaction between peptides with mutations at the 11th position (CLV3-S^{11th} and CLE41-H^{11th}) and receptors (CLV1 and TDR) by competition assays using [¹²⁵I]ASA-KIN photo-affinity labeling. The details are shown in p10-11, Line 224-236 and Supplementary Fig. 10. The conclusion is that the mutations at the 11th positions did not abolish the binding ability of the peptides to the receptors, consistent with the results from the MD simulations.

Reviewer #3 (Remarks to the Author):

This manuscript contains some interesting observations on the behaviour of several peptides related to CLV3. The authors initially looked at the difference between the activities of the CLV3 and CLE25 two peptides that have broadly similar spectrum of activity albeit with different activities on particular phenotypes such as root shortening. They were able to show that a particular synthetic peptide, that was essentially a hybrid between CLV3 and CLE25, also exhibited activity in similar manner to the TDIF peptide of CLE41 that normally exhibits a quite different, and in some cases opposite, set of activities to CLV3 and CLE25. It certainly is an interesting observation. The manuscript is clearly written and the conclusions sound. There appears to be no obvious problems with the data. Some of the error bars are quite large, but the appropriate statistical test appear to have been carried out. The respective activities of the hybrid "KIN" peptides requires the appropriate receptor kinase for either CLE41 or CLV3- like activity.

I am slightly unclear about is what this the central message of this paper. The authors show that they can generate what is essentially a novel peptide with novel bifunctional activity that is representative of 2 classes of CLE peptides. The claim is that this could be the route to entirely new classes of peptides with novel activities that could be used for different applications. It also sheds some light on the structure function relationship of CLE peptides.

Close inspection of CLV3 CLE25 and CLE41 show that in fact there is a lot of similarity between

CLV and CLE41 the central VPSGPDP is common to both that suggest that the 'class specificity lies outside this region. Of the 3 remaining amino acids at the N-terminus of CLE41 two are shared with CLE25. So the best hybrid of CLV3/CLE25 would have 9 of 10 identical amino acids at the N-terminus. Peptides, such as no. 12, that contain this sequence appear to have both kinds of activities. Incorporating K at position 2 to generate KIN enhances bifunctional activity. While this is an interesting observation it is unclear whether this is ever likely to be generally applicable as a useful means of generating novel peptides activities. There is no indication that this method might be somehow be applied more generally. Is this just a one off that results from the particular sequence similarities between CLE25/CLV3 and CLE41? In which case is it unlikely ever to be more useful than just controlling the expression of 2 peptides independently.

From the point of view about what this tells us about the structure function relationship between CLE peptides this appears to be a great starting point that raises some interesting questions, however few of these questions have been addressed in the manuscript and in this senses it is rather premature. It is also unclear whether the cryptic activity reveal has any biological meaning and is important as part of the CLE peptide signalling network.

A18. This is related to the comment from Reviewer 1, who summarized the novelty and implication of our findings (please also see our response A1). The central message of this manuscript is the successful engineering of plant peptides with an unnatural bioactivity, which has never reported or created previously. Although the applicability of our methodology to other peptide families is an open question, we believe that our findings will serve as a breakthrough that provides a new concept to engineer unnatural bioactivities by shuffling pre-existing plant peptide sequences. We agree that it is a future challenge to expand our technique and examine whether it is generally applicable to broader families of signaling peptides. We added some sentences in Discussion (p11-12), briefly describing these points.

Our approach, hybrid engineering, utilizes sequence variations existing in natural peptides. Such variations are found in almost all peptide families identified so far. During the course of molecular evolution, each family member has been shaped under selection pressures in its individual evolutionary path. For instance, in this study, we found that S11th of CLE41 plays a critical role to avoid unwanted misactivation of CLV1 signaling, which has never implicated by previous studies. As such, hybrid engineering could even serve as a methodology to explore “cryptic” roles for small variations in peptide sequences, some of which may be evolutionally avoided. It is plausible to imagine that this scenario, the selective avoidance of specific residues to prevent unwanted misactivation of evolutionarily close receptors

such as CLV1 and TDR for the CLE family, must also be common for evolution of other peptide hormone families. Thus, such a structural basis underlying the hybrid engineering implicates the general applicability of this methodology.

Successful engineering of KIN and CLE41-H^{11th} as bifunctional CLE peptides shows that there are multiple mutational routes to reach specific bioactivities using different pairs of natural peptides as starting materials. This result is conceptually consistent with the recent findings on the importance of promiscuous states of protein-protein interactions for their evolution (Sayou et al. 2014 *Science* 343, 645-648, or Aaker et al. 2015 *Cell* 163, 594-606), also implicating the general applicability of the hybrid engineering approach.

Collectively, our work will be a starting point for a number of future directions of peptide engineering in plants. For example, the species specificity of plant reproduction is caused by the species-specific variations of peptide hormones in some cases (Takeuchi and Higashiyama 2012 *PLoS Biol.* 10, e1001449). Hybrid engineering may be utilized to create an artificial “master key” to overcome reproductive barriers for the production of new beneficial plant/crop species.

REVIEWERS' COMMENTS:

Reviewer #1 (Remarks to the Author):

I was happy to see that the authors did an excellent job in improving the manuscript. They have addressed all my comments to my satisfaction. In my opinion there are no further points to raise that would not go beyond the scope of this paper (and have not been made in the first round of review).

I particularly commend their future outlook to how their approach of synthetic engineering of hybrid peptide might be used for further fundamental research but potentially also in applied approaches. I still think that the conceptual novelty does not only apply to receptors/ligands but to all protein-protein interactions.

Reviewer #3 (Remarks to the Author):

This manuscript does demonstrate that a novel peptide has bifunctional activity. It is unclear how widely applicable this approach is likely to be, but this can only be determined by further study. It certainly does extend the range of different approaches that can be used to engineer peptides with novel activities and is likely to get people to think more creatively about how they can engineer peptides with novel activities.